# Novel Targets and Advanced Therapies in Diffuse Large B Cell Lymphomas

**DOI:** 10.3390/cancers16122243

**Published:** 2024-06-17

**Authors:** Francesco D’Alò, Silvia Bellesi, Elena Maiolo, Eleonora Alma, Flaminia Bellisario, Rosalia Malafronte, Marcello Viscovo, Fabrizia Campana, Stefan Hohaus

**Affiliations:** 1Dipartimento di Scienze Radiologiche ed Ematologiche, Università Cattolica del Sacro Cuore, 00168 Rome, Italy; flaminia.bellisario@guest.policlinicogemelli.it (F.B.); rosalia.malafronte@guest.policlinicogemelli.it (R.M.); marcello.viscovo@unicatt.it (M.V.); fabrizia.campana@gmail.com (F.C.); stefan.hohaus@unicatt.it (S.H.); 2UOSD Malattie Linfoproliferative Extramidollari, Fondazione Policlinico Universitario Agostino Gemelli IRCCS, 00168 Rome, Italy; eleonora.alma@policlinicogemelli.it; 3UOC Servizio e DH di Ematologia, Fondazione Policlinico Universitario Agostino Gemelli IRCCS, 00168 Rome, Italy; silvia.bellesi@policlinicogemelli.it (S.B.); elena.maiolo@policlinicogemelli.it (E.M.)

**Keywords:** diffuse large B cell lymphoma, chimeric antigen receptor-T cells, bispecific antibodies, antibody–drug conjugates, BCL2-inhibitors, BTK-inhibitors, checkpoint inhibitors, XPO1-inhibitors

## Abstract

**Simple Summary:**

Diffuse large B cell lymphoma is the most common type of B cell neoplasm, and can be cured by traditional chemoimmunotherapy in 60–70% of cases. Relapsed/refractory patients usually have a dismal prognosis and most of them finally succumb to lymphoma. In the last decade, several new innovative therapies have been approved against DLBCL, targeting both surface antigens and the intracellular pathways of neoplastic B cells. Compared to standard chemotherapy, new monoclonal antibodies and engineered T cells have shown an increased response rate, progression-free survival and overall survival, but, on the other side of the coin, patients have been experiencing new kinds of toxicities, previously unknown, that clinicians have been learning to manage. Here, we review new drugs that are already approved or under investigation for DLBCL and the corresponding molecular target on lymphoma cells.

**Abstract:**

Since the introduction of rituximab in the late 1990s, significant progress has been made in advancing targeted therapies for B cell lymphomas, improving patients’ chance of being cured and clinicians’ therapeutic armamentarium. A better understanding of disease biology and pathogenic pathways, coupled with refinements in immunophenotypic and molecular diagnostics, have been instrumental in these achievements. While traditional chemotherapy remains fundamental in most cases, concerns surrounding chemorefractoriness and cumulative toxicities, particularly the depletion of the hemopoietic reserve, underscore the imperative for personalized treatment approaches. Integrating targeted agents, notably monoclonal antibodies, alongside chemotherapy has yielded heightened response rates and prolonged survival. A notable paradigm shift is underway with innovative-targeted therapies replacing cytotoxic drugs, challenging conventional salvage strategies like stem cell transplantation. This review examines the landscape of emerging targets for lymphoma cells and explores innovative therapies for diffuse large B cell lymphoma (DLBCL). From Chimeric Antigen Receptor-T cells to more potent monoclonal antibodies, antibody–drug conjugates, bispecific antibodies, checkpoint inhibitors, and small molecules targeting intracellular pathways, each modality offers promising avenues for therapeutic advancement. This review aims to furnish insights into their potential implications for the future of DLBCL treatment strategies.

## 1. Introduction

Diffuse large B cell lymphoma (DLBCL) is the most common subtype of mature B cell lymphoma, accounting for about 30% of all lymphoma cases in Western countries, with an estimated 150,000 new cases annually worldwide. The age-standardized incidence rate for DLBCL was 7.2 per 100,000 in the US. Incidence rates are rising in areas of historically low rates, such as Asia, while incidence rates in the Western hemisphere rose during the 20th century, but then appeared to plateau. DLBCL can arise de novo or as a transformation from a low-grade B cell lymphoma. The median age at diagnosis is in the mid-60s and identified rick factors include family history, several genetic susceptibility loci, viral infections (for instance EBV, HIV, HHV8, HBV, and HCV), solid-organ transplantation, autoimmune disorders, immunodeficiency, increased body mass index, pesticides, and ionizing radiation [1,2].

The cornerstone of treatment is the standard chemoimmunotherapy regimen R-CHOP, achieving cure rates of up to 60% for DLBCL patients. However, a significant portion of individuals experience relapsed or refractory disease, necessitating further interventions [1]. Historically, platinum-based regimens followed by autologous stem cell transplantation (ASCT) have served as the standard second-line treatment for relapsed/refractory DLBCL. Yet, in the post-rituximab era, this approach yields curative outcomes for only 20% of patients [3]. Particularly dismal are the prospects for primary refractory patients and those failing salvage chemotherapy or relapsing after ASCT, with median survival as short as 6.3 months [4]. For years, relapsed/refractory DLBCL has posed a great challenge, with efforts to surpass R-CHOP and ASCT proving largely futile. However, recent times have witnessed a notable shift, with the approval of several innovative therapies for DLBCL advancing these therapies from later to earlier lines of treatment. This transformation owes much to an enhanced understanding of the disease’s biology and pathogenetic pathways, accompanied by the technological advances in molecular diagnostics and immunophenotyping.

The evolving landscape of targeted therapies for DLBCL, particularly in the setting of R/R DLBCL, will inevitably change the financial burden of the management of these patients. A shift to an increase in pharmacy costs in the upcoming era of precision medicine can be foreseen. While this expansion of treatment options holds promise, it underscores the necessity for a more personalized approach to treatment. Such an approach seeks to optimize patient response and survival while mitigating adverse effects and rationalizing the escalating costs of medical care [5,6].

In this review, we explore the landscape of new drugs that are already approved or under investigation for DLBCL, elucidating the corresponding molecular targets in lymphoma cells. By tracing the trajectory from diagnostic processes to treatment selection, we aim to provide clinicians with valuable insights to navigate this evolving therapeutic paradigm.

## 2. Diagnostics for Clinical Decision Making in DLBCL

Our understanding of DLBCL has evolved to recognize its inherent heterogeneity in biology and drug sensitivity. In 2022, two updated classifications of mature B cell lymphomas were published: the 5th edition of the World Health Organization Classification of Haematolymphoid Tumours (WHO-HAEM5) and the International Consensus Classification of Mature Lymphoid Neoplasms (ICC) [7,8]. Within the WHO-HAEM5, DLBCL not otherwise specified (NOS) is classified in the family/class of large B cell lymphomas (LBCL), encompassing a spectrum of tumors characterized by aggressive clinical behavior. Differentiating DLBCL NOS from other LBCL types, particularly from DLBCL/high-grade B cell lymphomas (HGBL) with MYC/BCL2 rearrangements (double hit), typically involves morphological assessment, immunophenotyping, and fluorescence in situ hybridization (FISH) for BCL2, BCL6, and MYC rearrangements [9].

However, DLBCL NOS includes several molecular subtypes with distinct clinical outcomes and pathway dependencies. Gene expression profiling (GEP) in the early 2000s identified three DLBCL subtypes according to the ‘cell-of-origin’ (COO): germinal center B cell-like (GCB), activated B cell-like (ABC), and an ‘unclassified’ group [10,11]. While these subtypes exhibit differential responses to standard therapy and new drugs [12], the COO classification has limitations in predicting prognosis and treatment response. The nanostring approach, based on the expression of 20 genes, has emerged as a promising tool for molecular classification, capable of reproducing GEP data and identifying specific signatures, such as the double-hit signature (DHITsig), in FISH-negative cases [13,14]. 

Furthermore, genomic data integration has led to the identification of recurrent genetic alterations and genetic subtypes with distinct clinical outcomes and pathway dependencies [15,16]. The LymphGen algorithm was developed to determine the probability that a patient’s lymphoma belongs to one of seven genetic subtypes based on its genetic features and was able to classify 63.1% of DLBCL cases, while the remaining 36.9% were unclassified. Each subtype exhibits a distinct prognosis with standard R-CHOP therapy and varying sensitivity to different target therapies [17]. 

In the era of expanding immunotherapy strategies, the accurate immunophenotyping of biopsy samples is paramount for the subclassification and identification of target antigen expression, particularly in the relapse setting, where antigen downregulation may contribute to drug resistance [18]. A flow cytometry analysis of disaggregated tissue samples, alongside immunohistochemistry on formalin-fixed paraffin-embedded (FFPE) samples, enhances diagnostic accuracy and expedites diagnostic times, especially when using a limited material such as core needle biopsies [19].

## 3. CD19 as Target

CD19 is a type I transmembrane glycoprotein belonging to the immunoglobulin Ig superfamily, known for its specificity to the B cell lineage. It is expressed throughout the entire B cell maturation process and is typically lost in terminally differentiated plasma cells. Functionally, CD19 plays crucial roles in immune response, participating in antigen-independent B cell development and contributing to immunoglobulin-induced B cell activation. Notably, CD19 expression persists across various B cell malignancies, ranging from early B cell neoplasms like acute lymphoblastic leukemia to mature B cell lymphomas. The experimental evidence suggests that CD19 may collaborate with c-Myc in promoting lymphomagenesis, as demonstrated in Myc-transgenic mice [20,21]. Studies on CD19 antigen internalization upon antibody binding have generated conflicting results, and it is unclear whether this was due to differences in antibodies, cell types, or the co-expression of other molecules, such as CD21, on the surface of B cells [22]. 

### 3.1. Anti-CD19 Chimeric Antigen Receptor T Cells

Chimeric Antigen Receptor (CAR) T cell therapy targeting CD19 has demonstrated remarkable efficacy in treating aggressive and indolent B cell lymphomas, leading to FDA and EMA approval of three CD19-targeted CAR-T products for relapsed/refractory aggressive B cell lymphomas: axicabtagene ciloleucel (axi-cel; KTE-C19), tisagenlecleucel (tisa-cel; CTL019), and lisocabtagene maraleucel (liso-cel; JCAR017) [23,24,25]. These second-generation CAR T cell products incorporate an antigen-binding domain, hinge and transmembrane domains, a co-stimulatory domain (CD28 for axi-cel; 4-1BB for tisa-cel and liso-cel), and a T cell activation domain (CD3ζ) [23,24,25]. Liso-cel stands out due to its unique manufacturing methodology, which involves separately transducing isolated CD4 and CD8 T cell fractions from the apheresis product and sequentially infusing them in two equal target doses of CD4+ and CD8+ CAR+ T cell subsets. Despite their effectiveness, CAR T cell products are associated with class-specific adverse effects such as cytokine-release syndrome (CRS), immune effector cell-associated neurotoxicity syndrome (ICANS), hypogammaglobulinemia, and cytopenias, particularly B cell aplasia [26,27].

The pivotal trials investigating the efficacy of these CAR T cell products in patients with relapsed or refractory (R/R) large B cell lymphomas (LBCL) after two or more lines of therapy reported promising results (Table 1) [23,24,25].

Long-term follow-up data from these trials confirmed durable remissions, especially in patients achieving a CR [27,28,29]. With a median follow-up of 63.1 months, responses in the ZUMA1 trial were ongoing in 31% of patients, and the estimated 5-year OS rate was 42.6%, increasing to 64.4% in those patients who achieved a CR [28].

Real-world data, such as the retrospective French DESCAR-T registry study, have provided indirect evidence for differences in efficacy and toxicity between CAR-T products, aiding in treatment selection. Using propensity score matching (*n* = 418), the DESCAR-T investigators demonstrated the higher efficacy as well as greater toxicity of axi-cel compared to tisa-cel, with the best ORR/CR 80%/60% versus 66%/42%, 1-year OS/PFS 63.5%/46.6% versus 48.8%/33.2%, all-grade CRS 86.1% versus 75.6%, and all-grade ICANS 48.8% versus 22%, respectively [30]. 

In this context, liso-cel’s favorable safety profile, with a notably reduced incidence of severe CRS and ICANS, implies its potential suitability for a wider patient population. This may encompass individuals with characteristics such as renal dysfunction, mild left ventricular dysfunction, secondary CNS involvement, and advanced age who were included in the pivotal TRANSCEND trial [25].

Prospective randomized controlled trials (ZUMA-7, BELINDA, TRANSFORM) have further compared CAR T cell therapy to standard-of-care (SOC) high-dose chemotherapy with autologous stem cell transplant (ASCT) in refractory LBCL patients or those who relapsed within 12 months after first-line chemoimmunotherapy (Table 2) [31,32,33].

Of these trials, ZUMA-7 using axi-cel and TRANSFORM using liso-cel demonstrated the superior efficacy of CAR T cell therapy in terms of event-free survival (EFS), ORR, and CR rates compared to SOC [31,33], while the BELINDA trial using tisa-cel failed to meet its primary endpoint [32].

The exploration of CAR-T cell therapy in difficult-to-treat patient groups, such as those ineligible for ASCT, has shown promising results in open-label phase 2 trials (ALYCANTE and PILOT) using axi-cel and liso-cel, respectively. These trials reported high response rates (best CR 79% for axi-cel and 54% for liso-cel, with median PFS of 11.8 months and 9.3 months, respectively) and manageable toxicity profiles (grade > 3 CRS and ICANS occurred in 8.1% and 14.5% of axi-cel-treated patients, and 2% and 5% of liso-cel-treated patients [34,35].

Currently, CAR-T cell therapy has become the standard treatment for patients with relapsed/refractory DLBCL/HGBCL after two or more lines of treatment (axi-cel, tisa-cel, and liso-cel), as well as for patients with primary refractory or relapsed disease within 12 months after first-line therapy (only axi-cel and liso-cel) [36,37]. Nonetheless, several challenges persist, including CAR-T resistance mediated by CD19 downregulation, the diminished function of engineered lymphocytes due to prior treatments, and limited efficacy against the high disease tumor burden, and in the context of a heightened systemic inflammatory state, patients’ comorbidities and poor performance status, and logistical complexities. These logistical hurdles include the referral of patients to CAR-T centers, which can impact their socioeconomic status, and the extended manufacturing times, which are incompatible with rapidly progressive disease [18,26,27,38].

### 3.2. Anti-CD19 Antibody Tafasitamab

Tafasitamab, a humanized anti-CD19 monoclonal antibody, is augmented with an Fc domain designed to enhance its binding affinity to natural killer (NK) cells and macrophages, thereby facilitating antibody-dependent cellular cytotoxicity (ADCC) and antibody-dependent cellular phagocytosis (ADCP), alongside direct cytotoxicity [39]. Its efficacy is further potentiated by lenalidomide, which enhances ADCC, as evidenced by the predictive role of the number of circulating NK cells in response to this combination. Tafasitamab, in combination with lenalidomide, gained approval in the USA (July 2020) and Europe (August 2021) for treating adult patients with relapsed/refractory (R/R) diffuse large B cell lymphoma (DLBCL) who were ineligible for autologous stem cell transplant (ASCT) [40].

The pivotal study, L-MIND, a phase 2 trial, enrolled patients with R/R DLBCL ineligible for ASCT, including those with transformed indolent lymphomas with a limitation of no more than three prior lines of therapy [39]. Notably, patients with high-grade B cell lymphoma (HGBCL), prior exposure to anti-CD19 therapy or immunomodulatory drugs, central nervous system (CNS) involvement, or primary refractory disease were excluded. The treatment schedule and results are reported in Table 3. Neutropenia was the most common adverse event, occurring as grade ≥3 in 48% of patients. At 5-year follow-up, median DOR, PFS, and OS were 43.9 months, 11.6 months, and 33.5 months, respectively [41]. 

Data from tafasitamab–vlenalidomide-treated patients included in the L-MIND trial were compared to observational retrospective matched cohorts of patients treated in the real word with other systemic treatments approved for R/R DLBCL. In the RE-MIND study, tafasitamab + lenalidomide was superior to lenalidomide monotherapy in terms of ORR, CR, PFS, and OS [42]. Using propensity score-based 1:1 nearest-neighbor matching, the RE-MIND2 study showed improved an OS, PFS, DOR, ORR, and CR with tafasitamab–lenalidomide when compared to systemic pooled therapy, bendamustine + rituximab (BR), and rituximab + gemcitabine + oxaliplatin (R-GemOx) [43]. Similarly, a second comparative analysis in the RE-MIND2 study matched L-MIND patients with retrospective cohorts of patients treated with polatuzumab–vedotin + bendamustine + rituximab (Pola-BR), rituximab + lenalidomide (R2), and CAR-T therapies: in this analysis, tafasitamab–lenalidomide showed improved overall survival when compared to Pola-BR and R2 and comparable OS to CAR-T [44].

However, it is crucial to interpret these findings cautiously, considering the exclusion criteria of the L-MIND trial, which did not include primary refractory patients, those with a high cytogenetic risk, those refractory to more than four lines of therapy, or those with an Eastern Cooperative Oncology Group (ECOG) performance status >2. A real-world retrospective analysis of 178 R/R DLBCL patients treated with tafasitamab–lenalidomide demonstrated substantially different outcomes, highlighting the importance of patient selection for optimal therapeutic outcomes [45]. Ongoing investigations, such as the FIRST MIND phase 1b study in treatment-naïve DLBCL patients and the double-blind phase 3 Front-MIND (NCT04824092) study, aim to further elucidate the role of tafasitamab in DLBCL management [46].

**Table 3 cancers-16-02243-t003:** New monoclonal antibodies approved for treatment of DLBCL.

Name	Type of mAb	Target	Schedule of Treatment	Patients’ Population	Line of Treatment	Efficacy Data	Ref.
Tafasitamab	Fc-modified	CD19	Tafasitamab i.v. 12 mg/kg on day 1,8,15,22 for cycle 1–3 (28 days each) and on day 1 and 15 from cycle 4 until PD, in combination with lenalidomide 25 mg/day on days 1–21 of each 28-day cycle per 12 cycles	R/R DLBCL (including t-IL) ineligible or relapsed after ASCT ECOG 0–2 Excluded: DH/TH HGBCL and primary refractoy disease	2nd–4th lines of therapy	ORR 57.5% CR 41.3% mDOR not reached mPFS 11.6 months mOS 33.5 months	[39,41]
Loncastuximab-teserine	ADC	CD19	Loncastuximab i.v. 150 μg/kg on day 1 of 21-day cycle, for two cycles, then 75 μg/kg thereafter, for up to 1 year or until disease relapse or progression, unacceptable toxicity, or death	R/R DLBCL, DH/TH HGBCL, PMBCL, ECOG 0–2 Excluded bulky disease and t-IL	3rd line or more	ORR 48.3% CR 24.1% mDOR 10.3 months mPFS 4.9 months mOS 9.9 months	[47,48]
Glofitamab	BsAb	CD20/CD3 (2:1)	Obinutuzumab i.v. 1000 mg on day 1 of cycle 1, Glofitamab i.v. 2.5 mg on day 8, and 10 mg on day 15 of cycle 1, followed by 30 mg on day 1 of cycles 2 through 12 (cycles lasted 21 days)	R/R DLBCL, t-FL, HGBCL, PMBCL. ECOG 0–1	3rd line or more	ORR 52% CR 39% mDOR 18.4 months mPFS 4.9 months mOS 11.5 months	[49]
Epcoritamab	BsAb	CD20/CD3	Epcoritamab was administered subcutaneously weekly in cycles 1–3, once every 2 weeks during cycles 4–9 (days 1 and 15), and once every 4 weeks from cycle 10. Cycles lasted 28 days. Step-up dosing for 0.16 mg on day 1, 0.8 mg on day 8, and subsequent full 48 mg doses once on day 15 and beyond until disease progression or unacceptable toxicity.	R/R DLBCL, FL G3B, HGBCL, PMBCL. ECOG 0–2	3rd line or more	ORR 63.1% CR 38.9% mDOR 12 months mPFS 4.4 months mOS not reached	[50]
Polatuzimab-vedotin (PV)	ADC	CD79b	PV i.v. 1.8 mg/mq + BR every 21 days for 6 cycles	R/R DLBCL (excluded t-FL) ECOG 0–2	2nd line or more	ORR 41.5% CR 38.7% mDOR 9.5 months mPFS 6.6 months mOS 12.5 months	[51,52,53]
			PV i.v. 1.8 mg/mq + R-CHP every 21 days for 8 cycles (cycles 7 and 8 only PV + R) (double-blinded phase 3 trial, comparator arm R-CHOP)	DLBCL IPI 2–5 ECOG 0–2	1st line	ORR 85.5% CR 78% 2y PFS 76.7% 2y OS 88.7%	[54]

Abbreviations. mAb; monoclonal antibody; ADC: antibody–drug conjugate; PD: progression disease; R/R: relapsed/refractory; DLBCL: diffuse large B cell lymphoma; ASCT: autologous stem cell transplantation; t-FL: transformed follicular lymphoma; HGBCL: high-grade B cell lymphoma; DH/TH: double hit/triple hit; t-IL: transformed indolent lymphoma; PMBCL: primary mediastinal B cell lymphoma; FL G3B: follicular lymphoma grade 3B; ASCT: autologous stem cell transplantation; ORR: overall response rate; CR: complete remission; mDOR: median duration of response; mPFS: median progression-free survival; mOS: median overall survival; IPI: international prognostic index.

### 3.3. Anti-CD19 Antibody–Drug Conjugate Loncastuximab-Teserine

Loncastuximab–tesirine is an anti-CD19 antibody–drug conjugated through a cleavable linker to a potent pyrrolobenzodiazepine (PBD) dimer alkylating cytotoxin, SG3199. Upon binding to the CD19 antigen, Loncastuximab is internalized, the linker is cleaved, and PBD dimers are rapidly released. The PBD dimer forms persistent DNA crosslinks in the DNA minor groove, leading to tumor cell apoptosis [55]. LOTIS-2 was the pivotal study leading to FDA approval in April 2021, and EMA, in December 2022, conducted a multicenter phase 2 study for patients affected by DLBCL, HGBCL, and primary mediastinal B cell lymphoma (PMBCL), who relapsed or were refractory after two or more lines of treatment [47,56]. The results of LOTIS-2 are reported in Table 3. The median time to response was 41 days. Neutropenia (26%), thrombocytopenia (18%), and increased gamma-glutamyltransferase (GGT) 17% were the most common adverse events. Treatment-emergent adverse effects (TEAEs), likely related to the PDB, included oedema or effusion (31% cases), adverse events related to skin or nails (43%), and liver enzyme abnormalities (51%). All these events were generally mild to moderate in severity [47,48].

LOTIS-3 investigated Loncastuximab plus ibrutinib in R/R DLBCL and mantle cell lymphoma (MCL), achieving a CR rate of 34.3% and an ORR of 57.1% in DLBCL patients [57]. Ongoing trials (LOTIS-5, LOTIS-7) explore Loncastuximab combinations in R/R DLBCL [58,59].

## 4. Anti-CD20 Bispecific T Cell-Engaging Antibodies

The anti-CD20 antibody rituximab was the pioneering monoclonal antibody approved for the treatment of B cell malignancies. Its combination with standard chemotherapy has markedly enhanced outcomes for DLBCL patients, showing improvements in ORR, CR rate, PFS, and OS [60,61]. Despite the emergence of newer, humanized, and more potent antibodies such as ofatumumab and obinutuzumab, they have not surpassed the efficacy achieved by rituximab in combination with chemotherapy for DLBCL [62,63]. However, CD20 remains an appealing target for the development of advanced treatments against mature B cell malignancies.

CD20, a 33–37 kDa non-glycosylated transmembrane protein, belongs to the MS4A (membrane-spanning 4-domain family A) protein family. It serves as a B cell-specific marker, expressed from late pre-B lymphocytes onwards, with its expression lost in terminally differentiated plasma cells. In B cell malignancies, the expression level of CD20 varies widely based on the specific neoplasm, with DLBCL exhibiting the highest cell-surface expression. CD20 interacts with various other surface proteins on B cells, participating in microenvironment interactions and BCR-signaling [64].

T cell-engaging bispecific antibodies (BsAbs) targeting CD3 on T cells and CD20 on lymphoma B cells are emerging as a novel and promising immunotherapy for B cell lymphomas. These agents bind both target B cells and cytotoxic T cells, forming an immunological synapse that releases perforin and granzymes, leading to B cell destruction through lysis and cell death. BsAbs have demonstrated profound and enduring responses in relapsed/refractory (R/R) B cell lymphomas, with a favorable toxicity profile compared to CAR-T therapy.

Glofitamab, engineered with a novel 2:1 configuration of anti-CD20/CD3, has received approval for the treatment of adult patients with R/R DLBCL NOS, t-FL or primary mediastinal B cell lymphoma (PMBCL) who have received two or more lines of systemic therapy and are ineligible for, or have already received, CAR-T therapy [65]. In addition to inducing the expansion of pre-existing, intra-tumor, resident T cell populations, a preclinical study has shown that glofitamab promotes the recruitment of peripheral blood T cells [66]. The incidence of CRS in the phase 1 trial in R/R NHL was 50.3% (grade 3 or 4: 3.5%) and 1.2% of patients experienced grade 3 transient ICANS [67]. The phase 2 study enrolled 155 patients with relapsed R/R DLBCL who had received at least two previous lines of treatment. The results are shown in Table 3 and were consistent among the 52 patients who had previously received CAR-T therapy (35% of whom had a CR) [49]. The clinical development of Glofitamab as a monotherapy or in combination with standard therapies is ongoing in the setting of both R/R and treatment-naïve patients. 

Epcoritamab, an IgG1-bispecific, T cell-engaging antibody, is administered subcutaneously in 28-day cycles until disease progression or unacceptable toxicity for adult patients with R/R DLBCL NOS, including DLBCL arising from indolent lymphoma, and high-grade B cell lymphoma after ≥2 lines of systemic therapy [68]. The step-up dosage was set up in a phase 1/2 study in R/R NHL [69]. The dose-expansion cohort of the phase I/II study included 157 adults with R/R large B cell lymphoma and reported promising ORR and CR rates (Table 3), with manageable adverse events including CRS (grade 3 2.5%) and pyrexia. ICANS occurred in 6.4% of patients, with one fatal event. Similar responses were also observed in patients with primary efractory disease and in patients who received prior CAR T cell therapy [50]. 

Odronextamab, a hinge-stabilized, fully human, IgG4-based, CD20 × CD3-bispecific antibody, has demonstrated efficacy in R/R CD20-positive B cell malignancies with manageable CRS and ICANS. The phase 1 ELM-1 and phase 2 ELM-2 trials have shown notable ORR and CR rates, with consistent results across different dosing schedules [70,71]. The recommended step-up doses were 0.7/4/20 mg on days 1, 8, and 15 during cycle 1, followed by 160 mg on days 1, 8, and 15 of cycles 2–4. After C4, odronextamab maintenance treatment continued at 320 mg every 2 weeks until disease progression or unacceptable toxicity. At the final analysis of the phase 2 study including 141 patients, ORR and CR were 52% and 31%, respectively, and were consistent across high-risk subgroups. Safety was generally consistent with previous reports [71]. Odronextamab does not yet have marketing authorization in DLBCL.

While no comparative trials have been conducted among different anti-CD20/CD3 BsAbs, all have shown promising results in R/R DLBCL, including patients who relapsed after CAR-T therapy. They vary in treatment schedules, the routes of administration (intravenous vs. subcutaneous), and the duration of treatment (fixed vs. unlimited until disease progression or unacceptable toxicity). Epcoritamab and glofitamab have received FDA and EMA approval for R/R DLBCL after at least two lines of treatment, offering effective and accessible treatment options, which are particularly beneficial for patients with rapidly progressive disease where CAR-T therapy might not be feasible due to logistical challenges and the necessity of referral to specialized centers.

## 5. Anti-CD79b Antibody–Drug Conjugate Polatuzumab

The B cell receptor (BCR) is a complex comprising immunoglobulin (Ig) and a heterodimeric complex of two non-covalently associated proteins, Igα and Igβ, designated as CD79a and CD79b, respectively [72]. These proteins are pivotal for signal transduction via the BCR, a crucial pathway for B cell activation. Activation of this cascade leads to internalization of the CD79b-containing complex and its trafficking to late endosomes, thereby contributing to antigen presentation [73]. CD79b is expressed throughout B cell development, first appearing on the cell surface at the pre-B cell stage alongside the Ig heavy chain, but it is undetectable in terminally differentiated plasma cells [74]. The expression of CD79b varies among B cell neoplasms, being virtually absent in B-acute lymphoblastic leukemia (B-ALL) but widely expressed in mature B cell neoplasms, except for chronic lymphocytic leukemia (CLL) [75,76,77]. While CD79a’s expression in DLBCL is well-established, data on CD79b expression in diffuse large B cell lymphoma (DLBCL) are more limited and heterogeneous [77]. The near-exclusive expression of CD79b on B cells and B cell lymphomas, coupled with its internalization upon antibody binding, makes it an intriguing target for antibody–drug conjugates (ADCs) [78].

Polatuzumab vedotin (PV) is an ADC comprising an IgG1 monoclonal antibody targeting CD79b, covalently conjugated to the anti-mitotic cytotoxic agent monomethyl auristatin E (MMAE) via a cleavable linker. Upon binding to CD79b on the B cell surface, PV is internalized and MMAE is released into the cell, inhibiting division and inducing apoptosis [79]. PV has demonstrated activity against most DLBCL cell lines in vitro, irrespective of cell-of-origin subtype or CD79B mutation status [80]. In clinical trials, PV, alone or in combination with other therapies, showed an acceptable tolerability profile in DLBCL patients, with peripheral sensory neuropathy being the most common treatment-emergent adverse event leading to discontinuation, while the most serious TEAEs were diarrhea, lung infection, and lung disorder [51,52,53].

In 2019, the US FDA granted accelerated approval to PV, administered via intravenous infusion at a dosage of 1.8 mg/kg every 21 days for 6 cycles, in combination with bendamustine plus rituximab (PolaBR), for adults with R/R DLBCL who are ineligible for transplantation. The PolaBR combination significantly increased the CR rate (40.0%) compared to rituximab–bendamustine (BR) aloe (17.5%), with a reduced risk of death observed in clinical trials (Table 3). In the extension cohort, higher response rates and a longer median PFS and OS were observed in patients who received Pola-BR in second line of treatment, in patients who were not refractory to prior therapies, and in patients without primary refractory disease [52,53].

A subsequent analysis of the Polarix study (NCT03274492) suggests that adding Polatuzumab to standard first-line immunochemotherapy may improve PFS in DLBCL patients, particularly for those at high risk with an International Prognostic Index (IPI) of 3–5 and ABC cell of the origin subtype [54]. 

In a recent phase Ib trial (NCT02611323) involving patients heavily pre-treated with R/R NHL, a subset of DLBCL patients achieved complete or partial responses with acceptable safety when treated with PV in combination with venetoclax and anti-CD20 antibody [81]. Further trials are underway, evaluating PV in combination with various immunochemotherapy regimens in previously untreated DLBCL patients.

## 6. Anti-ROR1 Antibody–Drug Conjugate Zilovertamab–Vedotin

Receptor tyrosine kinase-like orphan receptor 1 (ROR1) is a transmembrane receptor belonging to the tyrosine kinase family (RTK). It comprises three distinct extracellular regions: an immunoglobulin (Ig)-like domain, a cysteine-rich domain (CRD), and a kringle domain (KNG). Intracellularly, it possesses a tyrosine kinase domain along with two serine/threonine-rich and proline-rich motifs [82,83]. Initially identified as an oncofetal protein expressed during embryogenesis, ROR1 mediates signaling from its ligand, Wnt5a, to drive embryonic stem cell proliferation in the development of various tissues, but its expression diminishes in normal adult tissues. However, ROR1 is selectively overexpressed in several human solid and hematological malignancies, potentially acting as a survival factor for tumor cells [82,83,84].

In hematological diseases, the strong expression of ROR1 has been described in CLL and mantle cell lymphoma (MCL), while moderate and heterogeneous expression has been noted in marginal zone lymphoma, DLBCL, and follicular lymphoma [85,86,87]. Immunohistochemical studies have associated ROR1 expression in DLBCL with poor prognosis subgroups, such as primary refractory DLBCL and Richter’s syndrome, while its expression is less frequent in de novo non-relapsed DLBCL [88]. The selective expression of ROR1 on malignant cells presents an attractive target for novel cancer immunotherapy, including monoclonal antibodies (mAbs) and small-molecule tyrosine kinase inhibitors, with a potentially lower rate of side effects [82,89].

Zilovertamab vedotin (ZV, MK-2140) is an antibody–drug conjugate targeting ROR1, conjugated to the anti-microtubule agent monomethyl auristatin E (MMAE) via a cathepsin-cleavable linker. Similar to other MMAE-containing antibody–drug conjugates, ZV binds to ROR1, leading to rapid internalization, trafficking to lysosomes, cleavage of the antibody–drug conjugate, and the release of MMAE, which inhibits cell division by blocking tubulin polymerization [90,91].

Initial results have shown that ZV has a tolerable safety profile and promising antitumor activity in patients with relapsed/refractory (R/R) NHL. In the first-in-human safety trial, phase 1 dose-escalation study waveLINE-001, 32 heavily pretreated NHL patients received ZV every 3 weeks until progression or unacceptable toxicity. Notably, three of the five DLBCL patients that were enrolled showed an objective tumor response (one partial and two complete responses). Adverse events included acute neutropenia and cumulative neuropathy, leading to a recommended ZV dose of 2.5 mg/kg every 3 weeks [91]. In the subsequent single-arm, open-label, phase 2 study, waveLINE-004 (NCT05144841), evaluating ZV monotherapy in 40 R/R DLBCL patients, preliminary results at a median follow-up of 6.0 months showed an ORR of 30% (10% CR, 20% PR) [92].

Currently, a phase 2/3 multicenter, open-label, randomized, active-control study of ZV in combination with standard of care (R-GemOX or BR) in R/R DLBCL patients (waveLINE-003) is underway [93]. Additionally, another multicenter, open-label, phase 2 dose escalation and confirmation, and efficacy expansion study (MK-2140-007) of ZV in combination with R-CHP in first-line DLBCL patients is ongoing [94].

ZV does not yet have yet marketing authorization in DLBCL.

## 7. Anti-CD30 Antibody–Drug Conjugate Brentuximab–Vedotin

CD30, a member of the tumor necrosis factor receptor superfamily, is expressed on a small subset of activated T CD4+ and CD8+ cells, as well as B lymphocytes [95]. It is notably overexpressed in various lymphoid neoplasms, with the highest expression observed in classical Hodgkin lymphoma (cHL), anaplastic large-cell lymphoma, and EBV-driven clonal lymphoproliferative disorders [96]. In the context of DLBCL, CD30 expression, typically evaluated by immunohistochemistry (IHC), varies in percentage and intensity.

For de novo EBV-negative DLBCL NOS, CD30 positivity, defined by a cut-off of 20% stained lymphoma cells, ranges from 10% to 21% across different studies, with a suggested trend toward better prognosis, especially in the GCB subgroup [97,98,99,100]. Conversely, utilizing a threshold of 5%, CD30 positivity is documented in up to 90% of cases of EBV-positive non-GC DLBCL, correlating with worse outcomes in this subset [101,102]. Additionally, CD30 is overexpressed in up to 80% of primary mediastinal large B cell lymphomas (PMBCL), 70% of post-transplant lymphoproliferative disorders (PTLD), and primary effusion lymphomas (PEL), suggesting a potential role for targeted therapy in these settings [103,104,105].

Early clinical trials with “naked” anti-CD30 antibodies yielded disappointing results [106]. Conversely, the ADC brentuximab vedotin (BV), comprising a humanized IgG1 anti-CD30 monoclonal antibody conjugated to the antimitotic agent MMAE via a cleavable linker, demonstrated significant efficacy in CD30-positive lymphomas [107]. Upon binding to CD30-positive cells, BV is internalized, leading to MMAE release within lysosomes, inhibiting tubulin formation and inducing cell apoptosis [107]. However, the correlation between CD30 expression levels and response rates is complex, with factors such as low expression or false-negative cases, “bystander killing,” and potential off-target immune modulatory effects influencing responses [108,109,110,111].

Several clinical studies have explored the role of BV in DLBCL treatment. BV in combination with Rituximab and CHP has shown promising activity as a first-line regimen for CD30+ DLBCL, including PMBCL and grey-zone lymphoma, with an impressive ORR of 100% (86% CR) in a phase I/II trial [112]. In a phase 2 study, single-agent BV demonstrated an ORR of 44%, with 16% CR and a median duration of response of 16.6 months in R/R CD30+ DLBCL [113]. Notably, there was no significant correlation between response and CD30 level, although all responding patients had quantifiable CD30 expression. BV has also shown efficacy in patients designated as CD30-negative by conventional IHC, with an ORR of 31% and CR rate of 12% [109]. With computer-assisted digital image analysis, 11 of 16 responders were designated as low-level positive (CD30 > 1%).

In a phase II trial of BV in patients with R/R EBV-positive, CD30-positive, mature T, NK, or B cell lymphoma, ORR was 48% and median PFS and OS were 6.2 and 15.7 months, respectively [114]. In a phase 1/dose expansion trial, Ward et al. evaluated the combination of BV 1.2 mg/kg every 21 days and the immunomodulatory drug Lenalidomide 20 mg administered continuously for a maximum of 16 cycles in 37 patients with R/R DLBCL. The ORR was 57% with a 35% CR. Median DOR was 13.1 months, median PFS was 10.2 months, and median OS was 14.3 months. Response rates were highest in 15 patients with CD30+ DLBCL (ORR 73% vs. 45%, CR 40 vs. 32%). The combination has a favorable safety profile, with the major side effect being neutropenia in 59%, requiring G-CSF in most cases [115]. The combination of BV with lenalidomide is being further explored in ECHELON-3, a phase 3 study comparing BV plus lenalidomide and rituximab versus placebo in R/R DLBCL patients, with PFS as a primary endpoint [116]. Additionally, in the CheckMate 436 study, the combination of BV and the checkpoint inhibitor Nivolumab demonstrated high antitumor activity, with an ORR of 70% and a CR of 43, as well as an acceptable safety profile in R/R PMBCL, as discussed in more detail in the next chapter [117].

BV does not yet have marketing authorization in DLBCL.

## 8. Checkpoint Inhibitors

T cell priming and activation are tightly regulated by central and peripheral checkpoints to modulate immune responses and prevent autoimmunity [118]. The peripheral checkpoint involves inhibitory programmed cell death protein 1 (PD-1), expressed on T cells, and its ligands, programmed cell death ligands 1 and 2 (PD-L1 and PD-L2), expressed on target cells, including cancer cells. Upon ligand binding, PD-1 recruits the protein tyrosine phosphatase SHP2 to the immunoreceptor complex, leading to the dephosphorylation of TCR-associated signaling molecules and attenuation of signaling, resulting in T cell exhaustion, decreased cytotoxicity, and the protection of target tissues against immune-mediated damage [119].

Cancer cells, including B cell lymphomas, can evade or suppress T cell-mediated cytotoxicity by exploiting this immune checkpoint [120]. However, there is considerable variability in PD-L1 expression measurements among studies. For example, using a cutoff of ≥30%, PD-L1 positivity ranges from 11% to 31% in B cell lymphomas, with over 90% positivity in cases of T cell/histiocyte-rich large B cell lymphoma (THRLBCL) [121,122]. PD-L1 expression has been associated with the ABC-type DLBCL subtype and inferior OS [123]. Moreover, PD-L1 is expressed in the tumor microenvironment of DLBCL, mainly on macrophages, and increased PD-1 levels in the peripheral blood of DLBCL patients have been linked to poorer prognosis [124].

About 10% of Primary Central Nervous System (PCNS) lymphomas and 36% to 100% of PMBCL are PD-L1-positive. Additionally, PD-L2 is highly expressed in up to 72% of PMBCL samples using an IHC cutoff of 20% [125]. Copy number gains in the 9p24.1 region containing the PD-L1/2 genes are rare in DLBCL but occur in 50% of PMBCL, primary testicular lymphoma (PTL), and PCNSL patients, leading to the increased expression of these proteins [126,127].

The PD-1/PD-L1 axis is particularly important for immune evasion in B cell lymphomas with a viral etiology, such as EBV- and HIV-associated lymphomas. For instance, PD-L1 positivity was reported in 100% of EBV + DLBCL cases and in 76% of EBV + THRLBCL cases [121,128].

Several monoclonal antibodies have been developed to block the proteins involved in downregulating immune responses, aiming to stimulate T cell-dependent cytotoxicity against tumors by reversing peripheral tolerance and T cell exhaustion [129]. Nivolumab and Pembrolizumab are humanized anti-PD1 IgG4 monoclonal antibodies that block the binding of PD-L1/PD-L2 to PD-1 due to their high affinity and specificity.

Nivolumab and pembrolizumab have acceptable safety profiles, with low rates of discontinuation resulting from adverse events, although close monitoring is warranted, especially for patients with a history of acute graft-versus-host-disease. Serious immune-mediated adverse events included pneumonitis, neutropenia, thrombocytopenia, hepatitis, rash, dyspnea, anemia, colitis, and others. 

While the use of PD-1/PD-L1 inhibitors as single agents has shown disappointing efficacy in R/R DLBCL, with CR rates ranging from 0% to 9% [130,131], or as a consolidation therapy following ASCT or CAR-T [132,133,134], they have shown promise in combination therapies. For instance, the combination of pembrolizumab with R-CHOP in untreated DLBCL patients yielded encouraging long-term outcomes [135,136]. In their study, Ho et al. [135] provided a long-term follow-up of a clinical trial involving pembrolizumab in combination with R-CHOP in 30 adults with untreated DLBCL. They found that at a median follow-up of 4.8 years, the 5-year PFS was 71% and the 5-year OS was 83%. Interestingly, immune-related adverse events occurred in 23% of patients, with 10% being grade 3/4. Notably, none of the patients with any PD-L1 expression experienced relapse, while two out of the four patients without PD-L1 expression did relapse. Similarly, Smith et al. [136] also reported a high rate of CR at 77% and a 2-year PFS of 83% among 30 patients with previously untreated DLBCL. They observed toxicity levels comparable to standard R-CHOP, with only two grade ≥3 immune-related adverse events. PD-L1 expression was associated with the non-GCB subtype and correlated with improved PFS and survival, suggesting that PD-L1 could serve as a biomarker to identify DLBCL patients who might benefit from this first-line treatment strategy.

The efficacy of PD-1 inhibition in DLBCL appears to vary depending on the specific disease subtypes. For instance, it may be more effective in certain subtypes such as PMBCL and primary large B cell of immune privileged sites (PLBLIPS) [137]. However, there are controversial results regarding the efficacy of nivolumab in primary central nervous system lymphoma (PCNSL). The CheckMate 647 study enrolled patients with relapsed/refractory (R/R) PCNSL and showed low objective response rates (ORR) of 6.4%. Similarly, for patients with primary testicular lymphoma (PTL), the ORR was 26.4%, and the responses were of very short duration [138].

In contrast, pembrolizumab showed promising results in PMBCL. In the phase 1b KEYNOTE-013 study, patients treated with pembrolizumab achieved an ORR of 41%, with a median duration of response (DOR) not reached after 11.3 months of follow-up [139]. In the phase 2, single-arm, KEYNOTE-170 study, PMBCL patients treated with pembrolizumab had an ORR of 41.5% and a CR rate of 20.8%. The median DOR was not reached, and the median PFS was 4.3 months, with a median OS of 22.3 months. Notably, none of the patients who achieved a CR progressed at the data cutoff. The median time to CR was 2.7 months, so pembrolizumab is not recommended for the treatment of patients with PMBCL who require urgent cytoreductive therapy [140].

As previously mentioned, the combination of nivolumab and brentuximab vedotin showed high antitumor activity in the phase 2 CheckMate 436 study in patients with R/R PMBCL patients after ASCT or after two lines of therapy who were not eligible for ASCT [117]. Thirty patients received nivolumab at a dose of 240 mg intravenously and BV at a dose of 1.8 mg/kg intravenously every 3 weeks until disease progression or unacceptable toxicity. At a median follow-up of 11.1 months, the ORR was 73%, the CMR rate was 43% and median DOR, PFS, and OS were not reached. Eleven responders underwent consolidation with transplant, including five autologous and six allogeneic transplants. However, the treatment was associated with some grade 3 to 4 treatment-related adverse events, observed in 53% of cases. The most common adverse events were neutropenia, thrombocytopenia, and peripheral neuropathy. However, these adverse events were manageable within the study protocol [117].

Pembrolizumab and Nivolumab have not yet achieved marketing authorization in DLBCL, while, in 2018, Pembrolizumab received FDA approval for the treatment of R/R PMBCL.

## 9. BCL-2 Inhibitors

Antiapoptotic B cell lymphoma-2 (BCL-2) family proteins play crucial roles in regulating the apoptotic process, contributing to the survival of cancer cells and enhancing tumor cell viability. BCL-2 homology domain 3 (BH3) mimetics, which act as inhibitors of antiapoptotic BCL-2 family proteins, have emerged as promising targeted therapies for various cancers. In diffuse large B cell lymphoma (DLBCL), BCL-2 is a key antiapoptotic molecule that is often overexpressed [141]. This overexpression is primarily attributed to the t(14;18) (q32;q21)/BCL2-IgH translocation in germinal center B cell (GCB) DLBCL and BCL2 locus amplification in the activated B cell (ABC) subtype [142,143].

Venetoclax, an orally administered BH3 mimetic, has garnered significant attention as a selective antagonist of the antiapoptotic protein BCL-2. Its therapeutic efficacy has been particularly notable in hematological malignancies [144]. While other small-molecule BCL-2 inhibitors are under investigation, venetoclax stands out due to its well-established safety profile and demonstrated efficacy in clinical studies [145]. 

In a pivotal phase 2 trial known as CAVALLI, the therapeutic efficacy and safety of combining venetoclax with R-CHOP chemotherapy were assessed in 206 patients diagnosed with diffuse large B cell lymphoma (DLBCL) and an international prognostic index (IPI) of 2–5 [146]. Patients received venetoclax orally at a dose of 800 mg on a noncontinuous schedule—days 4 to 10 of cycle 1 and days 1 to 10 of cycles 2 to 8—with each cycle lasting 21 days. The ORR was 83%, with 69% achieving a CR. The 2-year PFS rate stood at 80%, indicating promising outcomes. Adverse events (AEs) were manageable, although there was an increase in hematologic and infectious complications. Grade 3/4 AEs included cytopenias (neutropenia 68%, anemia 24%, thrombocytopenia 22%), febrile neutropenia (31%), and infections (23%) [146]. These findings suggest that the addition of venetoclax to R-CHOP does not compromise dose intensity but may heighten hematologic and infectious adverse effects. Notably, there was a suggestion of enhanced efficacy, particularly among patients with BCL2 overexpression observed via immunohistochemistry, underscoring the need for further investigation in randomized trials, which are currently underway [147].

Moreover, a recent phase I study explored the safety and preliminary efficacy of adding venetoclax to six cycles of dose-adjusted-R-EPOCH chemotherapy in 30 treatment-naive patients with aggressive B cell lymphoid neoplasms, including 30% with DLBCL [148]. While unacceptable dose-limiting toxicities were not observed at the maximum dose level of 800 mg for 10 days per cycle, the investigators identified 600 mg for 5 days as the recommended phase II dose when combined with R-EPOCH due to the improved overall tolerability and reduced duration of cytopenias, which facilitated the maintenance of chemotherapy dose intensity. Grade 3/4 hematologic complications were common, with neutropenia affecting 83% of patients, thrombocytopenia 70%, and anemia 60%. Additionally, 63% of patients experienced febrile neutropenia, highlighting the significant hematologic toxicity associated with this combination regimen [148].

Several ongoing clinical trials are assessing the efficacy of venetoclax in R/R DLBCL, either as a monotherapy or in combination with other agents. Initial studies indicate that venetoclax monotherapy demonstrates limited efficacy in R/R DLBCL. In a phase I study involving 34 patients, the ORR was 18%, with a rate CR rate of 12%, and a median PFS of 1 month [149]. 

Subsequent investigations explored the combination of venetoclax with bendamustine–rituximab (BR) in a phase 1b study comprising 22 patients with R/R DLBCL [150]. The combination demonstrated an ORR of 41%, a CRR of 14%, and a median PFS of 4 months. Patients received oral venetoclax at doses ranging from 50 to 1200 mg for 3, 7, or 28 consecutive days of each 28-day cycle, for up to six cycles. The recommended phase II dose for the venetoclax–BR combination was established as 800 mg daily continuously. Notably, 83% of patients experienced grade 3/4 adverse events, with neutropenia (60%) and febrile neutropenia (8%) being the most common. Additionally, venetoclax was investigated in combination with polatuzumab vedotin (PV) and rituximab in a phase Ib/II trial involving 57 patients with relapsed/refractory DLBCL [151]. A preliminary analysis revealed an ORR of 65%, including a CR in 38% of patients. With a median follow-up of 7 months, the median duration of response (DOR), PFS, and OS were 6, 4, and 11 months, respectively. Notably, 37% of patients experienced serious adverse events, and 79% experienced grade 3–4 adverse events.

Ongoing clinical trials continue to explore venetoclax in combination with other agents for R/R DLBCL, including obinutuzumab plus lenalidomide (NCT02992522), R-ICE chemotherapy (rituximab, ifosfamide, carboplatin, etoposide; NCT03064867), and Loncastuzumab (NCT05053659) [152,153,154]. These trials aim to further elucidate the efficacy and safety profile of additional venetoclax combinations in this challenging patient population.

Venetoclax does not have marketing authorization in DLBCL.

## 10. BTK Inhibitors

Bruton’s tyrosine kinase (BTK) is a critical player in the activation, proliferation, and survival of B lymphocytes, making it a significant target for drug development in B cell malignancies. Over the past decade, several BTK inhibitors (BTKi) have been developed, classified into covalent (e.g., Ibrutinib, Acalabrutinib, Zanubrutinib) and non-covalent inhibitors (e.g., Pirtobrutinib) based on their mechanisms of action and chemical structures. BTK expression has been detected in a significant number of cases of DLBCL, particularly in the ABC subtype [155]. Higher BTK expression has been associated with a more adverse IPI and poorer treatment outcomes. However, BTK inhibitors have shown limited single-agent activity in DLBCL.

In the context of first-line therapy, the phase 3 PHOENIX study assessed the efficacy of ibrutinib in combination with R-CHOP in non-GCB DLBCL [156]. The study included 838 patients randomized to receive either ibrutinib plus R-CHOP or placebo plus R-CHOP. The ORR and CR rate were similar in both arms, with an 89% ORR and 71.2% CR rate observed in the ibrutinib arm. Of particular note, in patients under 60 years of age, the addition of ibrutinib improved event-free survival (EFS), PFS, and OS. However, no such benefit was observed in older patients. This lack of clinical benefit in patients aged 60 years or older may be attributed to the significantly higher incidence of SAEs observed in the ibrutinib plus R-CHOP arm compared to the placebo plus R-CHOP arm (63.4% vs. 38.2%). Additionally, patients receiving ibrutinib plus R-CHOP were less likely to complete six cycles of R-CHOP, further contributing to the absence of clinical benefit in this age group. These findings underscore the importance of age-related considerations and potential toxicities when incorporating BTK inhibitors into first-line treatment regimens for DLBCL.

In a single-arm, phase 2, prospective study conducted at a single center, patients with newly diagnosed non-GCB DLBCL with extranodal involvement were treated with Zanubrutinib in combination with R-CHOP for six cycles, followed by two cycles of maintenance treatment with rituximab and zanubrutinib [157]. The ORR was 91.3%, with a CR rate of 82.6%. With a median follow-up of 16.7 months, the 1-year PFS and OS rates were reported to be 80.8% and 88.5%, respectively. Hematological AEs of grade 3 or higher included neutropenia (50%), thrombocytopenia (23.1%), and anemia (7.7%). Non-hematological AEs of grade 3 or higher included pulmonary infection (19.2%).

The SMART study investigated the efficacy of combining BTKi with lenalidomide and rituximab as a 21-day regimen for elderly patients with newly diagnosed DLBCL, administered for 6–8 cycles [158]. Alternatively, for patients unable to tolerate chemotherapy initially, a modified approach called SMART-START was employed, where BTKi combined with lenalidomide and rituximab was administered for the first 2–3 cycles, followed by the addition of chemotherapy [158]. A total of 31 patients were included. The ORR in the SMART group was 87.5%, with 62.5% achieving CR, while in the SMART-START group, the ORR was 92.3%, with a CR rate of 61.5%. With a median follow-up of 15.4 months, median PFS and OS were not reached. The 1-year PFS was 81% in the SMART group and 84% in the SMART-START group, while the 1-year OS was 89% in the SMART group and 91% in the SMART-START group. The most common grade 3–4 AE was neutropenia (8/31). Six patients experienced infections, including two cases of pneumonia, one of which was pulmonary fungal infection, two cases of febrile neutropenia, and one each of urinary tract infection and herpes zoster. Skin rash occurred in 11 patients, with three cases being grade 3–4, leading to lenalidomide interruption or dose reduction in two patients.

BTKi also demonstrates promising results in the setting of R/R DLBCL. A chemotherapy-free regimen was investigated in a phase 1b trial combining the BTKi ibrutinib with lenalidomide and rituximab [159]. The ORR was 44%, with a CR rate of 28%. Median PFS and overall OS were 6 and 10 months, respectively. The most commonly reported AEs included gastrointestinal events, myelosuppression, fatigue, hypokalemia, peripheral edema, and maculopapular rash. Serious AEs were observed in 56% of patients, including 24% with treatment-related serious AEs, with febrile neutropenia, atrial fibrillation, and dehydration being the most frequent.

Zanubrutinib was evaluated as monotherapy in a phase 2 study involving 41 patients with relapsed/refractory non-GCB DLBCL after a median of two prior treatment lines [160]. At a median follow-up of 6.8 months, the ORR was 29.3% with a CR rate of 17.1%. Median PFS and OS were 2.8 and 8.4 months, respectively. The most common treatment-emergent grade ≥3 AEs included neutropenia, pneumonia, abdominal issues, and death of unknown cause. Notably, there were no occurrences of atrial fibrillation, major hemorrhage, or tumor lysis syndrome.

In a multicenter, open-label, phase Ib study, acalabrutinib was investigated as a monotherapy in patients with relapsed/refractory non-GCB DLBCL [161]. The ORR was 24%, while in patients with the ABC subtype defined by the NanoString methodology, the ORR was 33%, with 22% achieving CR. Median PFS and OS were 1.9 months and 15.5 months, respectively. Treatment-emergent AEs occurred in 95% of patients, most commonly diarrhea and fatigue, with 29% experiencing grade 3/4 AEs.

Ongoing clinical trials are evaluating acalabrutinib (de novo DLBCL: NCT04546620, NCT02112526; R/R DLBCL: NCT03736616, NCT04189952) [162,163,164,165], Zanubrutinib (de novo DLBCL: NCT05189197; R/R DLBCL: NCT06033820, NCT03145064) [166,167,168], and Pirtobrutinib (R/R DLBCL: NCT03740529) [169].

BTKi has not yet obtained marketing authorization in DLBCL.

## 11. XPO1 Inhibitor Selinexor

Exportin 1 (XPO1) is a pivotal nucleo-cytoplasmic shuttling protein that orchestrates the export of proteins from the nucleus to the cytoplasm. In cancer, particularly in DLBCL, XPO1 plays a multifaceted role in tumorigenesis and therapy resistance. It mediates the functional inactivation of various tumor-suppressor proteins, including p53, p73, IkBκ, and FOXO, while also promoting the translation of oncoproteins such as c-Myc, Bcl-XL, Bcl2, Bcl6, survivin, and cyclin D1. By exporting oncoprotein mRNA complexes with eukaryotic translation initiation factor 4E (eIF4e) out of the nucleus, XPO1 contributes to the elevation of oncoprotein levels, fostering tumor growth and survival. Additionally, XPO1 enhances the resistance of lymphoma cells to chemotherapy by upregulating the translation of DNA repair proteins. Targeting XPO1 offers a promising therapeutic strategy by retaining multiple tumor suppressor proteins in the nucleus, preventing the translation of oncoproteins, inducing transient cell-cycle arrest, suppressing tumor growth, and promoting apoptosis. Importantly, XPO1 blockade has been shown to restore sensitivity to chemotherapy, presenting a potential avenue for overcoming treatment resistance in DLBCL [170,171,172]. Moreover, XPO1 is frequently overexpressed in DLBCL and correlates with poor prognosis, highlighting its significance as a therapeutic target in this aggressive lymphoma subtype [173].

Selinexor, an oral selective inhibitor of XPO1-mediated nuclear export, received FDA approval in June 2020 as single agent for treatment of R/R DLBCL patients who have undergone at least two lines of systemic therapy [174]. 

In the phase 2b SADAL study, a multicenter, multinational, open-label trial, heavily pretreated R/R DLBCL patients received 60 mg of selinexor orally on days 1 and 3 weekly until disease progression or unacceptable toxicity. Of the 127 evaluable patients, the ORR was 28%, with a CR rate of 12%. The median DOR was 9.3 months (23 months for CR and 4.4 months for PR), median PFS was 2.6 months, and median OS was 9.1 months. The most common grade 3–4 AE included thrombocytopenia (56%), neutropenia (25%), anemia (23%), fatigue (11%), hyponatremia (8%), and nausea (6%). Pyrexia (7%), pneumonia (5%), and sepsis (5%) were the most frequent serious AE [175]. 

A post-hoc analysis of the SADAL study by DLBCL subtypes revealed a higher ORR and longer median OS in DLBCL patients with normal c-MYC/BCL2 expression levels compared to those with c-MYC and BCL2 overexpression (46.2% vs. 14.8% and 13.7 vs. 5.1 months, respectively) [176].

Given its potential to overcome lymphoma chemoresistance, ongoing trials are investigating combinations of selinexor with other anticancer agents, aiming to enhance therapeutic outcomes in DLBCL patients.

Selinexor received marketing authorization in DLBCL from the FDA but not by EMA.

## 12. Future Developments in DLBCL

In addition to the previously described drugs that are approved or currently under investigation, many new, innovative therapeutic strategies are under development.

The evolution of CAR T cell therapy is ongoing and includes several new approaches to overcome the limits of conventional products, namely excessive toxicity and limited efficacy. Preliminary clinical data based on small and heterogenic patient samples have shown highly variable results and are insufficient to draw conclusions at present. Large-scale clinical trials are required [177]. Third-generation CAR-T cells carry two costimulatory domains, most frequently combining CD28 and 4-1BB, which may lead to the quicker expansion of T cells and more rapid elimination of tumor cells, but also causes CAR-T cells to remain longer in the host [177].

Dual-targeting CAR-T cells, which target two different antigens on lymphoma cells, such as CD19-CD20 or CD19-CD22, are showing promising results and can improve patients’ outcome by reducing relapses due to antigen loss [177]. Additionally, CAR-T products targeting antigens different from CD19, such as CD79 or ROR1, offer new valid targets and have shown robust antitumor activity in preclinical studies [178,179].

Armored CAR-T cells utilize several mechanisms to modulate the immune checkpoint and circumvent the inhibitory effect of the tumor microenvironment. These mainly include the following: (1) extracellular PD-1 fused to the intracellular CD28-activating domain, thus transforming PDL1—PD1 binding into an activating signal and increasing anti-tumor efficacy; (2) secretion of the PD-1 Fc fusion protein by CAR-T cells, which then binds to PD-L1, preventing its suppressive effects on T cells; (3) downregulation of inhibitory checkpoint molecules, such as the PD1 and ITIM (immunoreceptor tyrosine-based inhibitory motif) domains (TIGIT) or Hematopoietic Progenitor Kinase 1 (HPK1) [177].

T cells redirected for universal cytokine-mediated killing (TRUCKs) are next-generation CAR-T cells engineered to express and secrete certain cytokines (e.g., IL7 and CCL9). Following antigen recognition, in addition to cytotoxic activity, the engineered cells release selected cytokines, which promote the proliferation and survival of CAR-T cells and act as chemoattractants and enhancers of antitumor activity [177]. 

Switchable CAR-T cells incorporate several suicide mechanisms to deplete activated cells in case of treatment-associated toxicities, such as CRS and neurotoxicity. The surface antigen safety-switch strategy relies on surface proteins (e.g., EGFRt, HER2, RQR8) expressed by CAR-T cells, which can be targeted by a specific antibody to remove activated CAR-T cells via antibody-dependent cytotoxicity or complement-dependent cytotoxicity. The iCasp9 safety-switch approach induces the ablation of iCasp9-transduced CAR-T cells via the administration of AP1903, which leads to caspase-9 dimerization and activation of the apoptotic pathway. CAR-T cells carrying the *Herpes simplex virus thymidine kinase (HSV-TK) Mut 2* gene may be targeted by the prodrug ganciclovir, which is converted to ganciclovir–triphosphate, a competitive inhibitor of deoxyguanosine’s incorporation into DNA, and causes cell death by disrupting the replication process [177]

Universal allogenic “off-the-shelf” CAR-T and CAR-NK cells can solve the limitations of conventional CAR-T therapies, including the long manufacturing time, difficulties in mobilizing the appropriate quantity of T cells, and reduced T cell quality in heavily treated patients. Universal allogenic CAR-T can be obtained by removing major histocompatibility complex (MHC) and T cell receptor (TCR) molecules from donor-derived cells by genome-editing tools such as CRISPR/Cas9 or transcription activator-like (TAL) effector nuclease (TALEN) [177].

CAR-NK cells offer potential solutions to the challenges faced by CAR-T cell therapy because they identify target cells without MHC restrictions, do not cause graft-versus-host disease (GVHD), can be obtained from different sources, such as allogeneic pluripotent stem cells or umbilical cord blood, and do not induce CRS or ICANS. Like CAR-T cells, CAR-NK cells undergo genetic modification to express CARs designed to recognize specific antigens present on target cells, including costimulatory domains [180].

Among the new immunotherapies under investigation, there is a new category of BsAbs that engage NK cells. Some preclinical results targeting CD16 and CDp46 in NK cells and CD19 in tumor cells have already been reported [181].

The development of new ADC is not limited to new targets, but also explores new mechanisms of cytotoxic action for the payload. Microtubule-disrupting agents and DNA-targeting agents are currently the classes of payloads that are most widely used in ADC design. Targeted agents such as BCL2 inhibitors, spliceosome inhibitors, and transcription inhibitors targeting RNA polymerase II are among the payloads with different mechanisms of action [182]. Several new ADCs have shown promising results in monotherapy or in combination with rituximab against R/R DLBCL (Table 4).

Several new small molecules targeting intracellular pathways are currently under development for DLBCL. Cereblon E3–ubiquitin ligase modulators, termed CELMoDs, have anti-lymphoma activity via the degradation of the transcription factors Ikaros and Aiolos and upregulation of interferon stimulated genes, inducing tumor cell apoptosis and T/NK cell activation. Lenalidomide is already widely used in monotherapy or in combination against DLBCL. Several next-generation CELMoDs with an increased affinity for cereblon and specificity for Ikaros and Aiolos are in development, including avadomide, iberdomide, and golcadomide [187]. Other new interesting classes of drugs under investigation against DLBCL include mucosa-associated lymphoid tissue lymphoma translocation protein 1 (MALT1) inhibitors (e.g., safimaltib and ABBV-525), interleukin-1 receptor-associated kinase 4 (IRAK4) inhibitors (e.g., emavusertib) and macrophage checkpoint inhibitors (e.g., magrolimab, lemzoparlimab, and evorpacept) [187].

## 13. Conclusions

The treatment landscape of DLBCL is undergoing a significant transformation with the emergence of new therapies, prompting the publication of several recommendations and algorithms for their sequential incorporation into treatment regimens [188,189,190]. Notable advancements have been achieved in the relapsed/refractory setting, where CAR-T products (such as axi-cel and liso-cel) have begun to replace ASCT, particularly in patients that relapse early (within <12 months following the end of first-line therapy). CAR-T therapies, including tisa-cel, are considered the preferred option after two or more lines of treatment. Anti-CD20 bispecific antibodies (BsAbs) like epcoritamab and glofitamab offer “off-the-shelf” alternatives to CAR-T therapy after two lines of treatment, with reduced rates of grade 3 or more CRS and ICANS, and are more accessible, especially in peripheral centers. Loncastuximab–teserine and BsAbs are also commonly used after CAR-T failure or in patients ineligible for CAR-T therapy. Pola-BR (polatuzumab vedotin plus bendamustine and rituximab) and tafasitamab–lenalidomide have shown efficacy in R/R DLBCL patients ineligible for ASCT or CAR-T therapy, with better results observed when used in the second line of treatment rather than in more advanced lines and in relapsed disease rather than refractory disease. 

Many of these new therapies have demonstrated promising results, particularly when used in combination regimens and in specific DLBCL subtypes. However, most evidence comes from phase 2 clinical trials, emphasizing the need for future development to prioritize phase 3 randomized clinical trials and incorporate a thorough biologic lymphoma characterization and diagnostic accuracy to ensure the delivery of effective, safe, and personalized treatments to each patient.

## Figures and Tables

**Table 1 cancers-16-02243-t001:** The main studies on the use of CAR-T product in relapsed/refractory LBCL after two or more lines of therapy.

	ZUMA-1	JULIET	TRASCEND
Type of CAR-T	Axi-cel	Tisa-cel	Liso-cel
Number of patients	101/111 treated	115/167 treated	269/344 treated
Histologies included	DLBCL, PMBCL, tFL,	DLBCL, HGBCL, tFL,	DLBCL, PMBCL, HGBCL, tFL; FL G3B
Age, years	≥18	≥18	18–75
CAR-T cells infused	2 × 10^6^/kg	0.1 × 10^8^ to 6 × 10^8^	100 × 10^6^/kg
Type of conditioning	Cy-Flu	Cy-Flu	Cy-Flu
Median time from registration to CAR T infusion	NR	54 days	NR
Bridging therapy (% patients)	not allowed (only steroids)	allowed (90%)	allowed (59%)
CNS involvement	excluded	excluded	eligible
Best ORR/CR	83%/58%	53%/40%	73%/53%
EFS	38% (2 years)	-	-
PFS	-	31% (3 years)	41% (2 years)
OS	43% (5 years)	36% (3 years)	50% (2 years)
Any grade CRS/ICANS (%)	CRS 93% ICANS 64%	CRS 58% ICANS 21%	CRS 42% ICANS 30%
Grade ≥ 3 CRS/ICANS (%)	CRS 13% ICANS 28%	CRS 24% ICANS 14%	CRS 2% ICANS 10%

Abbreviations. CAR-T: chimeric antigen receptor T cells; DLBCL: diffuse large B cell lymphoma; t-FL: transformed follicular lymphoma; HGBCL: high-grade B cell lymphoma; PMBCL: primary mediastinal B cell lymphoma; FL G3B: follicular lymphoma grade 3B; Cy-Flu: cyclophosphamide-fludarabine; NR: not reported; CNS: central nervous system; ORR: overall response rate; CR: complete remission; EFS: event-free survival; PFS: progression-free survival; OS: overall survival; CRS: cytokine release syndrome; ICANS: immune effector cell-associated neurotoxicity syndrome.

**Table 2 cancers-16-02243-t002:** Randomized trials comparing CD19-CAR T versus ASCT in patients with DLBCL who were refractory to a precedent chemotherapy regimen or experienced disease progression within one year after the end of the first-line therapy.

	ZUMA-7	TRANSFORM	BELINDA
Type of CAR-T	Axi-cel	Liso-cel	Tisa-cel
Number of patients	359	184	322
Histologies included	DLBCL NOS, t-FL, HGBCL	DLBCL NOS, t-IL, HGBCL, Leg-type DLBCL, FL G3B, PMBCL	DLBCL NOS, t-IL, HGBCL, FL G3B, PMBCL
Age, years	≥18	≥18	18–75
CAR-T cells infused	2 × 10^6^ cells/kg	1 × 10^8^ cells	0.6–6 × 10^8^ cells
Median time from registration to CAR-T infusion	29 days	34 days	52 days
Bridging therapy (% patients)	Only steroids	Salvage chemotherapy (63%)	Salvage chemotherapy (97%)
Patients proceeding to CAR-T or ASCT	94% vs. 36%	97% vs. 47%	96% vs. 32%
CR (CAR-T vs. ASCT)	65% vs. 32%	74% vs. 43%	28% vs. 28%
EFS (CAR-T vs. ASCT)	8.3 vs. 2 months	n.r. vs. 2.4 months	3 vs. 3 months
PFS (CAR-T vs. ASCT)	14.7 vs. 3.7 months	n.r. vs. 6.2 months	n.a.
OS (months)	n.r. vs. 35.1 months	n.r. vs. 29.9 months	16.9 vs. 15.3 months
Any grade CRS/ICANS	92%/60%	49%/11%	61%/10%
Grade ≥ 3 CRS/ICANS	6%/21%	1%/4%	5%/2%

Abbreviations. CAR-T: chimeric antigen receptor-T cells; DLBCL NOS: diffuse large B cell lymphoma not otherwise specified; t-FL: transformed follicular lymphoma; HGBCL: high-grade B cell lymphoma; t-IL: transformed indolent lymphoma; PMBCL: primary mediastinal B cell lymphoma; FL G3B: follicular lymphoma grade 3B; ASCT: autologous stem cell transplantation; CR: complete remission; EFS: event-free survival; PFS: progression-free survival; OS: overall survival; CRS: cytokine release syndrome; ICANS: immune effector cell-associated neurotoxicity syndrome; n.r.: not reached; n.a. not available.

**Table 4 cancers-16-02243-t004:** New ADCs against DLBCL.

ADC	Target Antigen	Cytotoxic Payload	Clinical Data in DLBCL	Ref.
Coltuximab ravtansine (SAR3419)	CD19	maytansinoid DM4	In monotherapy, ORR 43.9% and CR 14.6% in R/R DLBCL	[183]
Denintuzumab Mafodotin (SGN-19A or SGN-CD19A)	CD19	MMAF	In monotherapy, ORR 33% and CR 23% in R/R DLBCL	[184]
Pinatuzumab Vedotin (DCDT2980S, RG-7593)	CD22	MMAF	In combination with rituximab, ORR 60% and CR 26% in R/R DLBCL	[185]
Naratuximab Emtansine (IMGN529)	CD37	maytansinoid DM1	In combination with rituximab, ORR 44.7% and CR 31.6% in R/R DLBCL	[186]

Abbreviations. MMAF: Monomethyl Auristatin F.

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
