# Peer review of "Novel Targets and Advanced Therapies in Diffuse Large B Cell Lymphomas"

_cancers, 2024, doi:10.3390/cancers16122243_

Round 1

Reviewer 1 Report

Comments and Suggestions for Authors

Thank you for the authors for this comprehensive review.

Some issues, however, should be resolved.

-Please add to table 1 explanations for abbreviations.

-There is no need to repeat the numbers from the table in the text, thus lines 135-139 are unnecessary and should be removed.

-The text should mention also drugs which are in clinical trials and if there are new mechanism of actions in pipeline. There are at least immunoconjugates with different toxine part, CAR-T-therapies, cereblon inhibitors, etc.

-If there are any treatment guidelines recommending treatment order across the listed new therapies, it would be valuable to refer to that.

-Please remove lines 382-385 and 427-8 to CAR-therapies (or a new paragraph of investigational drugs) 

 -The text should clearly state if (any) drug does not have yet marketing authorisation in DLBCL.

-The text is quite long and using tables showing the multiple numbers would absolutely ease the reader and thus is recommend and there are already in the article. The numbers should be removed from the text accordingly. Authors should consider also other information from the text to be changed to the table.

-Authors should consider again, what is the information, what is needed in this kind of text. Is it a lot of numbers or a little bite less numbers and more of evaluating the benefit of these treatments in clinical practice.

Comments on the Quality of English Language

-table 3: please correct glofitamaab. 

-some other minor typos noted.

Author Response

We modified the manuscript according to reviewers’ suggestions and tried to answer to each comments.

Reviewer 1

-Please add to table 1 explanations for abbreviations.

Answer: We added explanation for abbreviations in table 1, 2 and 3

-There is no need to repeat the numbers from the table in the text, thus lines 135-139 are unnecessary and should be removed.

Answer: lines 135-139 were removed

-The text should mention also drugs which are in clinical trials and if there are new mechanism of actions in pipeline. There are at least immunoconjugates with different toxine part, CAR-T-therapies, cereblon inhibitors, etc.

Answer: we added a new paragraph “Future developments in DLBCL” where we discussed new drugs under investigation.

-If there are any treatment guidelines recommending treatment order across the listed new therapies, it would be valuable to refer to that.

Answer: references to guidelines and therapeutic algorithms in DLBCL are reported in the conclusion section (ref. 188-190)

-Please remove lines 382-385 and 427-8 to CAR-therapies (or a new paragraph of investigational drugs)

Answer: lines 382-385 and 427-8 were removed and added to new paragraph about investigational drugs “Future developments in DLBCL”

 -The text should clearly state if (any) drug does not have yet marketing authorisation in DLBCL.

 Answer: we stated which drugs do not have marketing authorization in DLBCL

-The text is quite long and using tables showing the multiple numbers would absolutely ease the reader and thus is recommend and there are already in the article. The numbers should be removed from the text accordingly. Authors should consider also other information from the text to be changed to the table.

Answer: the table 3 summarized main clinical trials about approved drugs for DLBCL, so we removed numbers from the corresponding paragraphs and moved the table from the conclusion to inside the paragraph 3.

-Authors should consider again, what is the information, what is needed in this kind of text. Is it a lot of numbers or a little bite less numbers and more of evaluating the benefit of these treatments in clinical practice.

We thank the reviewer for this comment. In the text, we tried to critically describe recent advances in treatment of DLBCL that have changed current clinical practices. We removed some numbers from already validated and approved drugs, and described a little more interesting drugs under investigation.

Comments on the Quality of English Language

-table 3: please correct glofitamaab.

Answer: corrected

-some other minor typos noted.

Answer: we tried to identify them as much as possible

Thank you very much for your suggestions.

Reviewer 2 Report

Comments and Suggestions for Authors

   This manuscript comprehensively reviews novel CAR-T cell therapies for relapsed/refractory diffuse   large B-cell lymphoma. It details their mechanisms, clinical efficacy, and emerging challenges, offering valuable insights   for advancing treatment strategies and improving patient outcomes. The manuscript is well-written, interesting, and scientifically sound. 

Here are some minor changes to improve the manuscript.

1.     In the manuscript, write some information on the disease's epidemiology, incidence, trends, and burden of disease (the cost of the disease could also be included). To allowa readers from outside the area to have a better understanding

2.      Describe in one paragraph briefly the known or suspected risk factors. To allow a non-expert reader to have a global view.

3.     Please correct the reference number 16. References to "submitted" articles in ACS style are typically noted as "manuscript submitted" rather than just "submitted." ideally, it should also mention the journal to which the manuscript was submitted if known.

Here is how it should look with the journal included:

Bellesi, S.; Schiaffini, G.; Contegiacomo, A.; Maiolo, E.; Iacovelli, C.; Malafronte, R.; D’Innocenzo, S.; Alma, E.; Bellisario, F.; Viscovo, M.; et al. Enhancing lymphoma diagnosis on core needle biopsies: integrating immunohistochemistry with flow cytometry. Manuscript submitted to [Journal's Name].

If the journal name is not known, you can leave it as:

Bellesi, S.; Schiaffini, G.; Contegiacomo, A.; Maiolo, E.; Iacovelli, C.; Malafronte, R.; D’Innocenzo, S.; Alma, E.; Bellisario, F.; Viscovo, M.; et al. Enhancing lymphoma diagnosis on core needle biopsies: integrating immunohistochemistry with flow cytometry. Manuscript submitted.

If the manuscript is available from a repository such as ResearchGate or a specific webpage, you should include that information in the reference.

Bellesi, S.; Schiaffini, G.; Contegiacomo, A.; Maiolo, E.; Iacovelli, C.; Malafronte, R.; D’Innocenzo, S.; Alma, E.; Bellisario, F.; Viscovo, M.; et al. Enhancing lymphoma diagnosis on core needle biopsies: integrating immunohistochemistry with flow cytometry. Manuscript submitted. Available online: [URL] (accessed [date]).

4.     Please double-check in Table 1 that The dosage of tisa-cel in the JULIET trial is  5 × 10^8 cells per Kg and not per patient, or there is a typo.

5.     In the the discussion, speak a little more on emerging toxicities and the latest management strategies.

6.     In the discussion write a litle on future directions, discussing ongoing clinical trials, potential new CAR-T product development, and novel targets being explored for CAR-T therapy.

7.     In lines 696-699, there are references to several ongoing clinical trials, the trials should be included in the bibliography

For example, ClinicalTrials.gov. Study of Acalabrutinib in De Novo DLBCL. NCT04546620. Available online: https://clinicaltrials.gov/ct2/show/NCT04546620 (accessed May 20, 2024).

You should do this with all the clinical trials in the paragraph mentioned above..

Author Response

We modified the manuscript according to reviewers’ suggestions and tried to answer to each comments.

  1. In the manuscript, write some information on the disease's epidemiology, incidence, trends, and burden of disease (the cost of the disease could also be included). To allow readers from outside the area to have a better understanding

Answer: we included a quick reference about this issue in the introduction

  1. Describe in one paragraph briefly the known or suspected risk factors. To allow a non-expert reader to have a global view.

Answer: we included a quick reference about this issue in the introduction

  1. Please correct the reference number 16. References to "submitted" articles in ACS style are typically noted as "manuscript submitted" rather than just "submitted." ideally, it should also mention the journal to which the manuscript was submitted if known.

Answer: we added the journal where the manuscript was submitted and accepted for publication

  1. Please double-check in Table 1 that The dosage of tisa-cel in the JULIET trial is  5 × 10^8 cells per Kg and not per patient, or there is a typo.  

Answer: we corrected this datum, it was 0.1-6 x108 per patient

  1. In the discussion, speak a little more on emerging toxicities and the latest management strategies.

Answer: a brief reference to emerging toxicities was already present in each drug section, so we prefer to not make longer the already long manuscript.

  1. In the discussion write a little on future directions, discussing ongoing clinical trials, potential new CAR-T product development, and novel targets being explored for CAR-T therapy.

Answer: we added a paragraph “Future developments in DLBCL”

  1. In lines 696-699, there are references to several ongoing clinical trials, the trials should be included in the bibliography

For example, ClinicalTrials.gov. Study of Acalabrutinib in De Novo DLBCL. NCT04546620. Available online: https://clinicaltrials.gov/ct2/show/NCT04546620 (accessed May 20, 2024).

 You should do this with all the clinical trials in the paragraph mentioned above..

Answer: references to each clinical trial ongoing was added

 Thank you very much

 Francesco D’Alò

Round 2

Reviewer 2 Report

Comments and Suggestions for Authors

Dear authors the manuscript has improved significative congratulations for a well-done jobe